# *Caenorhabditis elegans* to Model the Capacity of Ascorbic Acid to Reduce Acute Nitrite Toxicity under Different Feed Conditions: Multivariate Analytics on Behavioral Imaging

**DOI:** 10.3390/ijerph18042068

**Published:** 2021-02-20

**Authors:** Samuel Verdu, Alberto J. Perez, Conrado Carrascosa, José M. Barat, Pau Talens, Raúl Grau

**Affiliations:** 1Departamento de Tecnología de Alimentos, Universidat Politècnica de València, 46022 Valencia, Spain; jmbarat@tal.upv.es (J.M.B.); pautalens@tal.upv.es (P.T.); rgraume@tal.upv.es (R.G.); 2Departamento de Informática de Sistemas y Computadores, Universidat Politècnica de València, 46022 Valencia, Spain; aperez@disca.upv.es; 3Department of Animal Pathology and Production, Bromatology and Food Technology, Faculty of Veterinary, Universidad de Las Palmas de Gran Canaria, Arucas, 35413 Las Palmas, Spain; conracarrascosa@gmail.com

**Keywords:** *C. elegans*, ascorbic acid, nitrite, behavior imaging, protective effect, multivariate analytics

## Abstract

Nitrocompounds are present in the environment and human diet and form part of vegetables and processed meat products as additives. These compounds are related to negative impacts on human and animal health. The protective effect of ascorbic acid has been demonstrated by some biological systems as regards several nitrocompounds. This work focused on studying the possibility of modeling this effect on nitrite toxicity with the model *Caenorhabditis elegans*. The three factors studied in this work were ascorbic acid concentration, nitrite exposure concentration, and presence/absence of food. The protective effect was evaluated by scoring lethality and its impact on behavior by means of multivariate statistical methods and imaging analytics. The effects of nitrite and the influence of food availability were evidenced. Apart from increasing lethality, nitrite had disruption effects on movements. All the observed symptoms reduced when ascorbic acid was administered, and it diminished lethality in all cases. Ascorbic acid maintained nematodes’ postural capacities. The results suggest that nitrites’ nonspecific toxicity in *C. elegans* can be mitigated by ascorbic acid, as previously evidenced in other biological systems. Thus, our results reveal the ability of *C. elegans* to reproduce the known protective effect of ascorbic acid against nitrite.

## 1. Introduction

Nitrogen is present in the environment in different forms, and its properties change throughout the nitrogen cycle [1]. Exposure to its derived compounds of dietary and environmental origins negatively impacts human and animal health [2]. Anion nitrates (NO_3_^−^) and nitrites (NO_2_^−^) are ubiquitous in the environment, transformed and nontransformed food products, industrial processes, and physiological systems. In humans, the main nitrate and nitrite sources are food, such as vegetables and processed meats, and are endogenously formed [3]. Nitrate and nitrite naturally form part of fruit and vegetables, mainly leaves, which represent an important fraction of a healthy human diet for their known beneficial health effects. They are widely used as food additives in products such as ham, sausages, bacon, and other processed meats because of their preservative and coloring properties. Unlike vegetables, processed meat is classified as carcinogenic to humans (group 1) by the International Agency for Research on Cancer (IARC-OMS, Agents Classified by the IARC Monographs, volumes 1–123). This is linked with the reduction of nitrate to nitrite in the gastrointestinal tract, which represents a potential precursor to form carcinogenic nitrosamine compounds, as well as the possibility of causing methemoglobinemia in older infants and children [4].

The difference between the above-mentioned food groups is that intake of dietary nitrate and nitrite from vegetables tends to increase nitrosation less due to the simultaneous presence of nitrosation inhibitors. These inhibitors are compounds with antioxidant activity, such as polyphenols and ascorbic acid, which avoid the endogenous conversion of nitrite into nitrosamine compounds or methemoglobinemia by enhancing reduction to nitric oxide (NO) [5,6].

This protective effect of ascorbic acid in relation to the toxicity of several N-compounds has been demonstrated by different biological systems both in vitro and in vivo. Arranz et al. [7] reported an in vitro study with human hepatocellular carcinoma cells (*HepG2*), where mechanisms by which ascorbic acid exerts its protective effect were detected, which appeared to be the interaction with enzyme systems by catalyzing the metabolic activation of N-nitrosamines and blocking the production of genotoxic intermediates. Ohsawa et al. [8] also evidenced the suppression of the genotoxicity of endogenously formed N-nitrocompounds by ascorbic acid in a study carried out with mice. Another example of these effects on humans comes with the study of Bahadoran et al. [9], who, over a 6-year follow-up study, observed how the modification of ascorbic acid impacts dietary nitrite and the incidence of type 2 diabetes.

One of the most powerful experimental in vivo models in toxicology is the nematode *Caenorhabditis elegans*. *C. elegans* presents high sensitivity to contaminants and other compounds contained in the medium. This property confers *C. elegans* the benchmark system condition for large toxicological studies. It offers several advantages as an animal model compared to other models, such as small size, short life, and rapid reproduction cycles, which confer easy handling [10]. This condition is based on the similitude to other biological systems such as mammals at genetic and physiological levels, in addition to having a simple nervous system whose alteration can be determined by studying movement patterns among other parameters. This means that modeling the effects of a given compound on several nematode parameters, such as lethality, locomotion properties, reproduction, growing kinetics, etc., has the potential to predict possible effects in complex biological systems such as higher animals [11]. Following this approach, several authors have already studied the effect of ascorbic acid on this nematode. Shibamura et al. [12] showed an increase in both the mean and maximum life span when ascorbic acid was administered in several ways, e.g., encapsulated in liposomes. Bakaev and Lyudmila [13] also demonstrated prolonged nematode longevity by ascorbic acid exposure, but only when supplementation was initiated in early life stages. This work intends to explore the possibility of modeling the protective effect of ascorbic acid on nitrite toxicity with the nematode *C. elegans*.

Thus, the objective of this work is to study the impact of ascorbic acid on acute nitrite toxicity under different feed conditions using the biological model *C. elegans* by studying lethality and behavior features.

## 2. Material and Methods

### 2.1. C. elegans Preparation and Exposure Protocols

Wild-type *Caenorhabditis elegans* Bristol strain N2 was employed as the biological model. The nematode population was reproduced on nematode growth medium (NGM) plates that had been seeded with *Escherichia coli* OP50 at 20 °C [14]. The nematode population was synchronized by a standard bleach method [15]. Age-synchronized L4 nematodes were used to perform the experiment. An assay was run for 48 h, divided into two 24-hour phases. The first 24-hour period was pre-exposure to ascorbic acid (AsA) solution, while the next one involved exposure to NO_2_^−^ solution to carry out the lethality assay. The exposure parameters were based on Sonane et al. [16] with modifications.

Exposures were achieved in liquid K-medium (32 mM KCl, 51 mM NaCl, pH = 7 ± 0.2) following the protocols of Zhou et al. [17]. Pre-exposure to ascorbic acid (L-Ascorbic acid 99%, Sigma-Aldrich^®^) was done in 3.4 mL well plates (Corning^®^ 3738 Costar^®^ 24-Well Flat Bottom) at two different concentrations (5 mM and 10 mM), and included a control group (K-medium without ascorbic acid). Exposure time was based on Sonane et al. [16] with modifications, and concentrations were established following the ranges of previous studies that had studied the ascorbic acid effect on *C. elegans* [13]. Note that, although ascorbic acid can be biosynthesized by this nematode, it was assumed to be part of the natural metabolism and, therefore, as nonvariable.

After 24 h of ascorbic acid exposure, the nematodes were washed twice with K-medium and singly transferred to sterile 96-microwell plates, where exposure to nitrite (NaNO_2_ ≥ 97%, Sigma-Aldrich^®^) solutions was done during the second 24-hour period. Exposure to nitrite was done at four different concentrations (1000, 3000, 6000, and 10,000 mg/L), including a control group (K-medium without nitrites). To avoid modifying compound concentration by means of external factors, the plates were kept away from light at 20 °C at all times.

The presence (P)/absence (A) of food was also tested in both phases to evaluate the influence of food availability on the studied toxicological parameters. The aim was to study the dependence between food intake and compound intake during exposure [18]. The used food was *Escherichia coli* OP50 administered as a concentrated pellet, regardless of the other compounds. The food presence combinations were modulated based on the above-mentioned assay phases (24 h ascorbic acid/24 h nitrite) by testing three different cases: absence/absence (AA); presence/absence (PA); presence/presence (PP).

After completing the procedure, the nematodes were counted as live or dead under a stereomicroscope by gently probing with a platinum needle. Responses were plotted on curves expressed as mortality percentages to calculate *LC_50_*. The combination of factors resulted in 30 different treatments. Each treatment was tested with at least 60 nematodes.

### 2.2. Artificial Vision System and Tracking Video Capturing

The behavior of nematodes was studied based on the features extracted from the analyses of tracking the videos captured during the lethality assays. Two 1-minute videos were captured per nematode during exposure to nitrite at 1 h and 24 h. These two time points were selected to collect representative information with the minimum impact on the experiment. The results were obtained as the summation of both captures. The employed device was a web cam controlled by a computer (Logitech C920 HD Pro Webcam, Logitech Europe S.A., Switzerland), which was placed perpendicularly at 3.5 cm to a white-polarized light source generated by a TFT-LCD 7” screen (NHD-7.0 OLED, Newhaven Display Intl, IL, USA). Both elements were placed inside a light-isolated chamber (20 × 20 × 20 cm). Videos were captured on 300 µL microwells (V96 MicroWell^®^ Clear Plates), into which nematodes were individually inserted for the 24-hour NO_2_ exposures. The microwell plates were placed between the camera and light source for the recording process. Videos were captured at 30 fps in the RGB format and at a resolution of 1080 × 720 pixels.

### 2.3. Video Preprocessing

Videos were processed to later acquire analyzable data about nematode behavior during exposure. The process was based on two steps: (1) the nematodes were segmented from video images; (2) the behavior descriptors of the segmented nematodes were obtained from the registered movements. In the first step, an automatic procedure was run to localize the nematode, whose behavior was recorded as a particle in movement. The second step was based on noting any descriptive features of movement [19]. The Appendix A summarizes the specimen segmentation procedure.

### 2.4. Behavioral Analysis

The behavioral analysis was performed based on the properties of movement during the locomotion of each recorded nematode by discretizing movements and extracting image descriptors for the later analysis. Discretization of movements facilitates the study of nematode postures while tracking. It has been used widely for this purpose because it reports information about systemic alterations and contributes to quantify disruptive effects from a given agent on multiple organisms, including humans [20]. The behavioral analysis was divided into two different blocks of descriptors, which collected information about locomotion metrics (based on displacements) and postural dynamics (based on worm shape). These two blocks were organized as follows:

Locomotion metrics (Block A): The objective of this block of descriptors was to collect displacement information in terms of the amount of movement per treatment, regardless of how it was done in postural dynamics terms. The descriptors used to collect information were
*M*: summation of total displacement in pixels*t*: time used to complete *M**V*: velocity of movement during the recorded minute, calculated from *M* and *t**A_m_*: effective area of movement, calculated as
(1)$$Am=\left(P_{x}^{max}-P_{x}^{min}\right)\times \left(P_{y}^{max}-P_{y}^{min}\right)$$
where *P^max^* and *P^min^* represent the maximum and minimum values of the pixel coordinates in the image during the video, respectively, calculated for the *x* axis and the *y* axis. From both the results, the effective area is theoretically calculated as a rectangle.

*ACT*: percentage of activity. It represents the amount of nematodes with recorded movement from the total, expressed as a percentage*M/A_m_*: the relation between M and A_m_, interpreted as the amount of movement per unit of effective area*M/ACT*: the relation between M and Am, interpreted as the amount of movement per active nematode.

Postural dynamics (Block B): This block focused on collecting information about locomotion features in posture terms within each frame, regardless of the amount of movement or displacement. The descriptor collecting this information was the maximum nematode length within each frame, which is a direct function of the body bend level.

The obtained time series of worm length percentages (Figure 1A) was transformed to obtain comparable dimensionality, which represented the features of the postures recorded during the set time. This transformation was based on generating a histogram of frequencies for the worm length values in an attempt to condensate and normalize the descriptor of temporal behavior. Worm lengths were discretized in 100 values from 0% to 100% worm length (Figure 1B). The Appendix A collected the procedure to obtain that histogram. Figure 1 is an example of a time series of raw worm length and the histogram obtained after the above-mentioned data transformation. These histograms were called postural spectra and showed the frequency of each % worm length in terms of % time during the tracking video. These spectra were processed as a multivariable data matrix from a statistical point of view. Following previous studies about the fundamentals of *C. elegans* behavior regarding locomotion properties, postural spectra were divided into two zones (*runs* and *turns*) according to recorded length. The ranges for each zone were 0–50% worm length for *turns* (Figure 2, gray zone) and 50–100% worm length for *runs* [21,22].

### 2.5. Statistical Analysis

Image descriptors were explored and compared after applying multivariable statistical procedures to reduce dataset dimensionality. To this end, the multivariate unsupervised principal component analysis (PCA) method was used to explore the interdependency of the defined descriptors and their load for explaining the variance obtained from all treatments. This method provides a way to simultaneously study the relation between descriptors and treatments and to facilitate the global effect of the compounds being observed. It was used to analyze information about both locomotion metrics (Block A) and postural dynamics (Block B). To examine in depth the links among the observed effects, the relation of the postural dynamics with locomotion metrics and mortality was tested by the multivariate supervised partial least-square-regression (PLS-R) method. It was used to test the linear dependences between the descriptors of the postural spectra (processed as a multivariate matrix after transforming them into artificial variables called latent variables *LV*, which are linear combinations from the original ones) with each descriptor from the locomotion metrics and mortality individually. These procedures were performed with the PLS Toolbox, 6.3 (Eigenvector Research Inc., Wenatchee, WA, USA), a toolbox extension in the Matlab 7.6 computational environment (The Mathworks, Natick, MA, USA).

## 3. Results

### 3.1. Lethality Assay

The results of the dose–response studies are plotted in Figure 2. The minimum LC_50_ for the treatment with no ascorbic acid pre-exposure (Figure 2A) was for treatment *AA* (~2900 mg/L), followed by *PP* (~3800 mg/L), while *PA* was the highest (~6.500 mg/L). This last one did not surpass 80% mortality at 10,000 mg/L, while *AA* and *PP* obtained mortalities over 90% at 6000 mg/L. The results showed that nematodes better resist nitrites than rats. In rats, 121 mg/kg bw is assumed, which means 27.83 mg nitrite for a standard rat weighing 230 g. When comparing these data with the complete amount of total nitrite in LC_50_ for treatment *AA* (0.87 mg), the relation between both masses revealed greater resistance by nematodes, a fact that should be taken into account when compared to human toxicity. That is crucial because the CONTAM Panel of EFSA (European Food Safety Authority) endorses an acceptable daily intake (ADI) for humans of 0.07 mg/kg bw/day, which was calculated from the NOAEL of rats, 6.7 mg/kg bw/day, by applying an uncertainty factor of 100 [23].

The presence of food had a significant impact on mortality. When food was included at the same time that ascorbic acid (PA), mortality reduced. However, when food was also administered in the nitrite exposure phase (PP), mortality increased, which revealed a similar response to the total absence of food (AA) (Figure 2B,C). Nitrite toxicity diminished after ascorbic acid exposure, and *LC_50_* increased in all cases. The treatments with 5 mM ascorbic acid presented more than double *LC_50_* for *AA* and *PP* compared to those treatments without ascorbic acid. For *PP* (Figure 2B), ascorbic acid allowed mortality to remain close to zero until 3000 mg/L nitrite in the same way as *PA*, which did not pass 10% mortality until concentrations exceeded 6000 mg/L nitrite. The treatments with 10 mM of ascorbic acid with food presented a similar response to 5 mM, although mortality was not observed until 3000 mg/L nitrites in this case. This is a significant effect because the absence of food led to mortalities between 10–20% nematodes at 1000 mg/L nitrites, regardless of the ascorbic acid pre-treatment.

The influence of food intake with the toxicity of a given compound has been previously reported, and revealed the dependence of the ingested dose of that compound with increased food intake. Similarly, Höss et al. [24] reported results indicating that the main uptake route of cadmium for nematodes was governed by food ingestion rather than by passive diffusion via the worm’s body surface. Toxic exposure upon food intake increases the toxic effect because nematodes introduce both components into their organism. Thus, increased food intake is accompanied by increasing nitrite. It is known that the presence of bacteria stimulates food intake [25]. Therefore with *PP*, the exposure route also went through internal zones, in addition to the worm’s body surface, as in the other two cases. Following this phenomenon, the observed effect of ascorbic acid fitted the results. The ascorbic acid exposures in the presence of food led to higher survival rates, as observed in both *PA* and *PP*. Therefore, although mortality lowered in all cases, the sharpest drop in them all was for *PA*.

### 3.2. Behavior Analysis

#### 3.2.1. Block A: Locomotion Metrics

The part of behavior corresponding to the study of locomotion metrics (Block A) was analyzed by a PCA to simultaneously study the relation of the descriptors from this block and treatments. Figure 3 shows three biplots of the same space of variance. Series were separated based on the ascorbic acid concentration to improve viewing the series. *PC1* and *PC2* collected 70.1% and 19.2% captured variance, respectively. All the descriptors had a considerable load to explain the variance captured by *PC1*. They were placed in the positive zone of this axis. Thus, the descriptors *V*, *M*, and *t* could be considered the variables with the most load variance when interpreting the position of treatments across *PC1*. Moreover, *PC2* presented a high load from the *M/A_m_* and *M/ACT* ratios in the positive zone, while *A_m_* and *ACT* did so in the negative one. The variance generated by the nitrite concentration was collected across *PC1* for all treatments. The control cases were grouped in the positive zone, from which the position of the other treatments tended to move toward negative values, mainly for the cases with 6000 mg/L nitrite.

The displacement to the negative zones meant a reduction in nematodes’ displacement capacities due to nitrite effects. An overall reduction in the percentage of activity (*ACT*) and in the other descriptors was evidenced, following the increase in concentration across that axis. This effect was mitigated when nematodes were pre-exposed to ascorbic acid. These cases were placed in the negative zone of *PC2*, which was closely related to *ACT* and *A_m_*. All cases, except for the exposures at 6000 mg/L nitrite, were collected in the positive zone of *PC1*. Their near position implied a similar behavior to the cases without nitrites. The main effect of ascorbic acid involved an increase in *ACT* and *A_m_* in all cases, where the presence of food did not bring about any conclusive differences between treatments. This result was in accordance with the lower mortality observed in the lethality assay, which explained the increase in *ACT*.

#### 3.2.2. Block B: Postural Dynamics

The properties of movements were studied from the Block B data. Figure 4 shows three plots as an example of the raw spectra obtained from three treatments based on the above-cited differences in Figure 3. Figure 4A represents the postural spectra for controls *AA*, *PA*, and *PP* (with no ascorbic acid and nitrite), Figure 4B shows the same cases affected by 3000mg/L nitrite, while Figure 4C depicts the last ones after 24 h of 5 mM ascorbic acid exposure.

The control cases had similar spectra. Most of the frequency went to the zones of the spectra of the *runs* postures, while the *turns* and extreme zone of *runs* obtained minimum frequencies. The main difference went to *AA*, which presented a peak in the *turns* zone (≤50% worm length). This peak may be related to the absence of food because when that stimulus is absent, the probability of producing turns significantly increases and improves the probability of finding food across the medium [21]. This was not the case of *PA*, which could mean that food in the first exposure phase could palliate those swimming patterns. Figure 4B shows that the main change was for *AA*, which significantly reduced in the *turns* zone, with the interval increasing to 80–100%, and pertained to the maximum length in the *runs* zone. Conversely, *PA* and *PP* increased in the *turns* zone. These modifications are not observed in Figure 4C. Ascorbic acid seemed to keep the postural spectra quite stable compared to the control treatments despite the nitrite exposure for all cases.

All the postural spectra were used as a battery of discrete variables in a multivariate matrix to optimize the simultaneous study of the collected information using *PCA*. Figure 5A–C depicts three biplots from the same space of variance using the spectra from all treatments. The series was separated based on the ascorbic acid concentration to improve the viewing of series. *PC1* collected 65.3% of variance, with 14.58% for *PC2*. The obtained space of variance showed that the postural spectra were divided into three groups (circumferences). These groups corresponded to the *runs* and *turns* zones of the spectrum (Figure 1). The *turns* postures were grouped in the positive zones of *PC1* and *PC2* (red circumference), while the interval of *runs* was divided into two different groups: the first corresponded to the approximate range of 50–85% (black circumference), placed in the positive zone of *PC1* and in the negative one of *PC2*. The zone from approximately 85–100% (green circumference) represented the maximum nematode length and was taken as a subgroup of postures called *M-runs* to improve the analysis. These postures represented the fraction of *runs* with the minimum body bends.

The observed grouping allowed us to know the properties of movements for all treatments simultaneously based on the location of each treatment in this space of variance. Next, the variance produced by nitrite concentration was collected by *PC1*. The increment in nitrite displaced the cases from the positive zone of *PC1*, where the controls were placed, to the negative one (Figure 5A). This displacement took place for all treatments, but differences were observed for the presence of food. The maximum impact was observed for the *AA* cases. Both *AA* 1000 and 3000 mg/L nitrite were placed in the *M-runs* zone, *PA* and *PP* presented an intermediate displacement toward the *M-runs* and *turns* zones, while the control cases were in the positive one near the *runs* zone.

The treatment with 5 mM ascorbic acid (Figure 5B) resulted in a displacement of *AA* 1000 and 3000 mg/L nitrite near the *runs* zone and the *AA* control. The *PA* treatments showed no remarkable differences. Likewise, the treatment with 10 mM ascorbic acid magnified the grouping of all the treatments near the control treatments. To visualize these behaviors on some real tracks, three examples of real trajectories are included in Figure 6A–C. These trajectories correspond to the *AA* control (Figure 5A), *AA* 3000 mg/L nitrite (5B), and *AA* 5 mM ascorbic acid/3000 mg/L nitrite (5C). Each point that formed tracks is colored according to the posture zone of the nematode within each frame following the groupings in the biplot. The trajectories from the control and the case treated with ascorbic acid (Figure 4A,C, respectively) presented similar characteristics in terms of the temporal fractions of each postural spectra zone: *turns* 14.5–10.6%; *runs* 81–85.2%; *M-runs* 4.5–4.3%. Conversely, *AA* 3000 mg/L nitrite presented modifications mainly between *turns* and *M-runs*, and showed quite a linear track. *Turns* dropped to 2.3%, while *M-runs* increased to 10.8%, which was almost double compared to the other two cases.

This observed reduction in the amplitude of body bends in *C. elegans* has been previously related to the disruption of motor control functions associated with the synaptic contacts of neurons and muscle rows. Li et al. [26] described this tendency of a lowering bending degree of nematodes when increasing acrylamide concentrations in the exposure medium. This behavior was also dose-dependent in that case and closely associated with lower survival. This behavior shortened the time in turning events, which fitted the above-mentioned possible disruptions to motor function. The lower probability of turning events has been related to an increasing compound gradient during the automatic monitoring of behavior in chemotaxis assays [22]. The differences brought about by the presence of food showed how *AA* was the most affected (Figure 5A). So despite *PP* having a higher nitrite intake and a similar *LC_50_* to *PP*, food availability conversely caused fewer alterations to the postural dynamics. These results evidenced the effect of ascorbic acid on the alterations in the postural dynamics and showed that increasing ascorbic acid concentrations helps to maintain nematodes near the control treatments.

Finally, PLS-R studies were carried out to study the relation between the Block B information with mortality and Block A. The postural dynamics spectra (Block B) were used as a multivariate data matrix (X), which individually correlated with mortality and the descriptors from Block A (Y). It provided information about the relation between the zones from the postural spectra and those descriptors based on the loadings of the generated *LV*. Table 1 shows the obtained *R*^2^ of calibration (Appendix A). The values went up to 0.85 for them all.

The maximum correlation was for *M* (0.93) and the minimum one for *M/A_m_* (0.86), with 0.92 for mortality. The results showed us a relation between the metrics of locomotion and the postural dynamics, which was altered by both nitrite and ascorbic acid. As the PLS-R method tests linear dependences between both data blocks, the load of frequency for each posture in the explanation of the observed correlations can be studied. Figure 7 showed the loadings from each *LV1* for *M*, *A_m_*, and mortality.

The loadings for *M* (Figure 7, black line) showed a high positive load in the *runs* zone. A positive peak around 80% worm length was observed, while negative charges were noted in the *turns* zone and *M-runs* of around 99–100% worm length. This evidenced the direct relation between the summation of total displacement and the postures from the 70–90% worm length interval, which meant a longer distance when a nematode was moving by adopting these postures. This matched the negative load of the *turns* zone for this descriptor, which meant a reduction in *M* when the time in the *turns* postures increased. Moreover, the loading spectrum of *A_m_* presented only positive loads, which appeared to be quite homogeneously distributed across worm lengths. Non-highlighted peaks were observed for this descriptor, but a depression in the *M-turns* appeared. The positive loads of the *runs* zones meant exactly the same in this case; an effective area of movement increased with correct *runs* postures but, unlike *M*, the increase in *turns* frequency also contributed to the rise in *A_m_*. This effect was caused by the probability of change in direction increasing when turns occurred, followed by the expansion of the effective area of movements in different directions in the image coordinates. Mortality presented a parallel behavior to *A_m_*, but in the negative zone of the axis. All the postures were inversely proportional to two main peaks occurring in both the *turns* and *runs* zones, which falls in line with that observed in the aforementioned descriptors. Thus the mortality produced by nitrite seemed to be related principally to a reduction in the *runs* zone of around 70–90% worm length and in the *turns* zone, which meant a disruption to normal nematode behavior in postural frequency terms, followed by a higher death probability.

## 4. Discussion

The observed reduction in nitrite toxicity by means of ascorbic acid was in accordance with the studies carried out with other previously reported biological systems and N-compounds. It evidenced the capacity of *C. elegans* as a biological model to reproduce these phenomena. Some of those cases were human cell lineages, mice, and humans, as mentioned above [7,9]. With HepG2 cells, ascorbic acid also protects these human cells from oxidative DNA damage induced by several nitrosamines, such as N-nitrosodimethylamine, N-nitrosopyrrolidine, and N-nitrosodibutylamin [7]

By taking into account the mechanisms observed in other organisms, the main toxic effects described for nitrite could be applied to interpret the observed nematode results. Nitrite is highly reactive with blood hemoglobin and converts it into methemoglobin and, thus, reduces blood’s oxygen-carrying capacity for mice and humans [27]. The effect on oxygen transport was not equal because *C. elegans* belongs to a genus with no circulatory system and hemoglobin. However, this animal uses pseudocoelomic fluid as a medium to signal between organs to transfer nutrients as proteins and oxygen. Although the exact balancing of oxygen levels in this fluid is not well understood, proteins with homologies to myoglobin or hemoglobin have been found in the nematode pseudocoelom, and they potentially act as carriers of oxygen, nitric oxide, or other gases [28,29]. So nitrite might interfere with that mechanism by disrupting either the capture or transport of oxygen in nematodes and by producing hypoxia-like conditions. That effect could explain both the mortality and reduction of *M*, *V*, *A_m_*, *ACT*, etc. This reduction could take place because only a small amount of oxygen is available. Hence, the alteration to muscle functions would generate displacement difficulties. This behavioral response due to limitations in oxygen availability has been previously described in *C. elegans*. In a medium with enough oxygen levels, nematodes create a random pattern of movement. This movement involves frequent turns and changes in direction during long runs of forward motion, but the frequency with which nematodes change direction is less when they detect low-oxygen conditions [30]. The nematode response occurs to modify motion by reducing turns, and a roaming form of displacement has been described that allows them to escape from a low-oxygen environment [31]. Such behavior matches the observed modification of the postural spectra during tracking, with fewer turns and changes in direction, while *M-runs* increased following a rising nitrite concentration (Figure 6).

Nitrosamine toxic effects were also taken into account, but conditions may not be optimum to generate nitrosamine, although secondary or tertiary amines with a nitrosating agent like nitrite could be simultaneously presented. The main reason was that the pH outside and inside nematodes was not favorable for the necessary reactions. Optimum nitrosamine formation in the human stomach has been reported at pH = 2–3.5 after 3–6 h [32], but the pH of the nematode intestine was estimated to lie between 4–5 [33]. We cannot guarantee that a certain amount of nitrosamines was not generated but, given the experimental conditions, their load within the observed toxic effects was probably lower.

Moreover, nitrite modifications could be altered by the presence of ascorbic acid. The endogenous conversion of nitrite into the main derived products, such as nitric oxide or nitrosamine, has been studied, and some potential factors such as the presence of antioxidants, particularly ascorbic acid, seems to intervene in the result. The main involved mechanism is to avoid nitrosamine production because ascorbic acid enhances the reduction of nitrite to nitric oxide. Ascorbic acid usually reacts with N_2_O_3_, H_2_NO^+^, and NO_X_ at rates higher than the corresponding nitrosation rates for amides and, thus, inhibits the nitrosation of nitrocompounds. This property has been exploited to reduce nitrosamine formation in foods matrices and in vivo systems [5,6].

This phenomenon could be related to the observed protection effect of ascorbic acid, which contributes to reduce the presence of nitrite and then its effects. At the same time, the reduction in mortality and the normalization of behavioral properties could be linked with nitric oxide formation. It is known that this compound directly modulates the activities of diverse proteins by binding to their active centers [34,35] and also protects from oxidative stress [34] and cardiovascular, immune, and neuronal systems [35,36]. These protective effects of nitric oxide on *C. elegans* have been previously reported by Gusarov et al. [36]. These authors showed how by feeding *C. elegans* with *Bacilli*, the bacterium that produces nitric oxide itself, and the longevity and stress resistance of nematodes were significantly enhanced. Thus, a reduction in nitrite by ascorbic acid could also contribute to the global protection effect based on the observed results.

The dose–response curves showed a positive dependence between mortality and nitrite concentration. The presence of food before and during exposure had a significant effect on *LC*_50_. The nematodes’ food status influenced resistance during nitrite exposure, and those nematodes with no food intake before being exposed displayed higher mortality. The simultaneous presence of food with nitrite increased *LC*_50_ regardless of food status. This evidenced the link of the ingested dose and food intake for this compound. Duration of movements was shorter, and the distance, velocity, and area traveled by nematodes also obtained lower values. The nitrite concentration presented an inverse relation with the amplitude of body bends and turn movements, which evidenced a disruption effect on normal behavior. Pre-exposure to ascorbic acid lowered *LC*_50_ in all cases, and up to 3-fold in some cases. The effect on behavior was an increase in the duration of movements, velocity, and effective area up to concentrations of 3000 mg/L nitrite. The impact on the postural dynamics maintained the nematodes’ bending capacity. The linear relation between the studied descriptors and nitrite effects was evidenced, which suggests the nonspecific toxicity of that compound on *C. elegans,* which could be mitigated by ascorbic acid, as previously evidenced in other biological systems. Our results showed the aptitude of *C. elegans* to reproduce the known protective effect of ascorbic acid against nitrite. Thus, using this biological model allows the effects of those compounds in an animal model to be studied with a short experimentation time to draw conclusions that complement and complete larger studies in more complex biological systems such as rats, humans, pigs, etc. This offers the possibility of undertaking more in-depth studies about the effect of ascorbic acid on chronic nitrite exposure to obtain more information about different aspects such as life span, fertility, health span, gene expression, etc.

## Figures and Tables

**Figure 1 ijerph-18-02068-f001:**
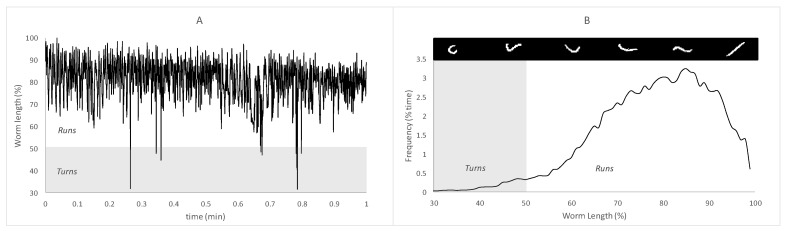
Data transformation from Block B. (**A**): raw time series of % worm length; (**B**): histogram of Figure 1. Gray and white zones indicate the *turns* and *runs* zones of the spectrum, respectively.

**Figure 2 ijerph-18-02068-f002:**
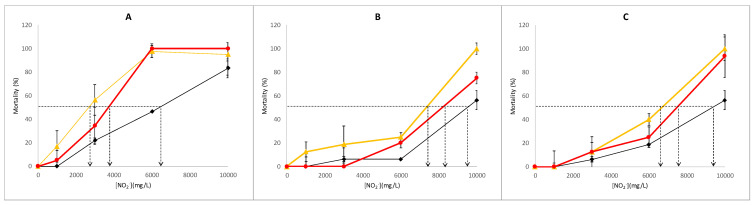
The results of the lethality assays. (**A**): dose–response curves with no ascorbic acid pre-exposure group; (**B**): the dose–response curves of the 5 mM ascorbic acid pre-exposure group; (**C**): the dose–response curves of the 10 mM ascorbic acid pre-exposure group. Dotted lines mark LC_50_ for each treatment. AA: yellow triangles; PA: black diamond; PP: red circles.

**Figure 3 ijerph-18-02068-f003:**
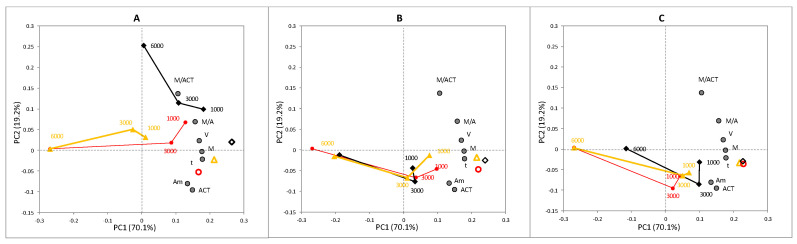
Biplot of the space of variance generated in the principal component analysis (PCA) study from the locomotion metrics data (Block A). (**A**): the score distribution of the treatments with the no ascorbic acid pre-exposure group; (**B**): the score distribution of the treatments with the 5 mM ascorbic acid pre-exposure group; (**C**): the score distribution of the treatments with the 5–10 mM ascorbic acid pre-exposure group. AA: yellow triangles; PA: black diamond; PP: red circles. Empty symbols represent the control cases (without nitrite) exposure group.

**Figure 4 ijerph-18-02068-f004:**
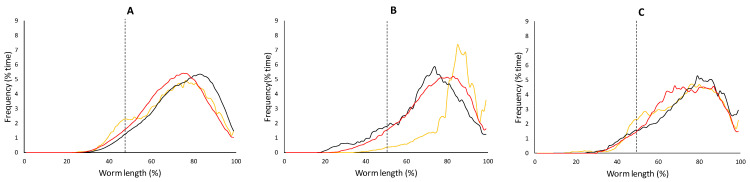
Example of the effect of nitrite and ascorbic acid on the raw postural spectra. (**A**): nematodes with no nitrite exposure; (**B**): nematodes at 3000 mg/L nitrite; (**C**): nematodes 5 mM ascorbic acid/3000 mg/L nitrite. Dotted lines mark the *turns* (0–50 %) and *runs* (50–100%) zones. AA: yellow; PA black; PP: red.

**Figure 5 ijerph-18-02068-f005:**
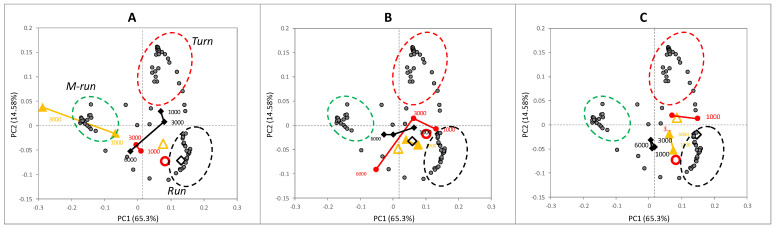
Biplots of the space of variance generated in the PCA study from the postural dynamics data (Block B). (**A**): the score distribution of the treatments with no ascorbic acid pre-exposure; (**B**): the score distribution of the treatments with the 5 mM ascorbic acid pre-exposure; (**C**): the score distribution of the treatments with the 10 mM ascorbic acid pre-exposure. AA: yellow triangles; PA: black diamond; PP: red circles. Empty symbols represent the control cases (without nitrite). Dotted circumferences indicate the accumulations of % worm length from the postural spectra corresponding to turn (red), run (black) and M-run (green).

**Figure 6 ijerph-18-02068-f006:**
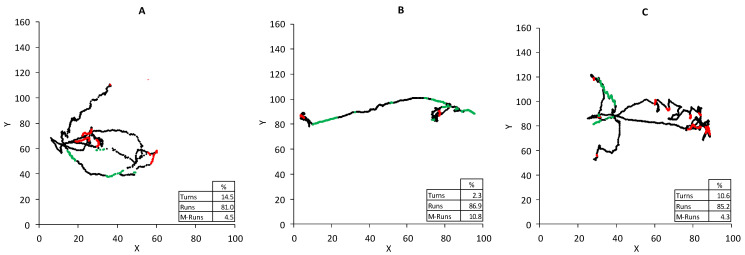
Example of the effect of nitrite and ascorbic acid on the real trajectories of nematodes. (**A**): AA control; (**B**): AA 3000 mg/L nitrite; (**C**): AA 5 mM ascorbic acid/3000 mg/L nitrite. Colors indicate the zones of the postural spectra corresponding to turns (red), runs (black), and M-runs (green). The legends show the % of each zone during tracking.

**Figure 7 ijerph-18-02068-f007:**
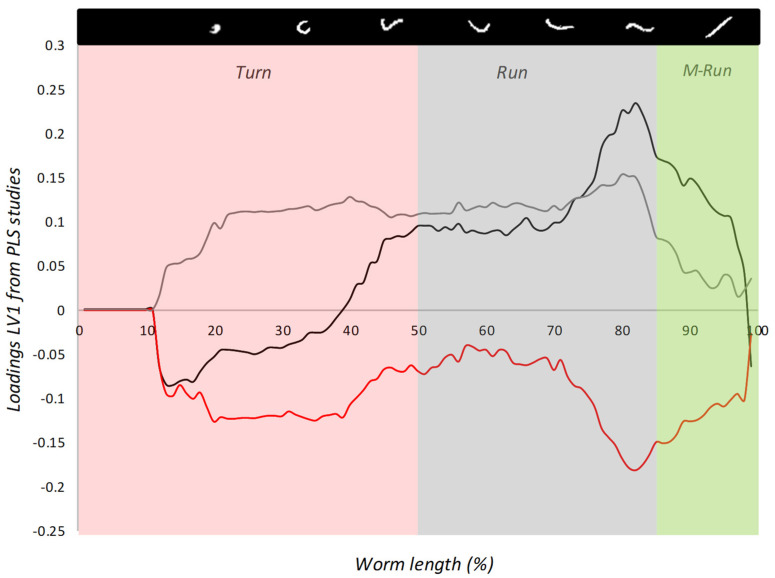
Loadings from latent variable 1 (*LV1*) of mortality, *M*, and *A_m_*. Lines show the load of variance from each zone of the postural spectra in the explanation of the calibration coefficients: *M*: black line; *A_m_*: gray line; mortality: red line. Divisions of the % worm length axis indicate the zones of the postural spectra corresponding to *turn* (red), *run* (black), and *M-run* (green). The legends show the % of each zone during tracking.

**Table 1 ijerph-18-02068-t001:** Regression studies of Block A and mortality with Block B.

Descriptors of Block-A	*R* ^2^
***M***	0.93
***t***	0.91
***V***	0.92
***A_m_***	0.91
***ACT***	0.92
***M/A_m_***	0.86
***M/ACT***	0.89
***Mortality***	0.92

*R*^2^: calibration coefficient from the partial least-square regression (PLS-R) studies; *M*: total displacement; *t*: time; *V*: velocity of movement; *A_m_*: effective area of movement; *ACT*: percentage of activity; *M/A_m_*: *M* and *A_m_*; *M/ACT* ratio, *M* and *ACT* ratio.

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
