# Peer review of "Caenorhabditis elegans to Model the Capacity of Ascorbic Acid to Reduce Acute Nitrite Toxicity under Different Feed Conditions: Multivariate Analytics on Behavioral Imaging"

_ijerph, 2021, doi:10.3390/ijerph18042068_

Round 1

Reviewer 1 Report

The manuscript can be accepted in the present form

Reviewer 2 Report

The manuscript has been significantly improved and is acceptable in the current form.

This manuscript is a resubmission of an earlier submission. The following is a list of the peer review reports and author responses from that submission.

Round 1

Reviewer 1 Report

Caenorhabditis elegans to model the capacity of ascorbic acid to reduce acute toxicity of nitrite under different feed conditions: multivariate analytics on behavioral imaging overall is an unclear manuscript with extremely confusing narrative. Below are some examples illustrated. Furthermore, the objective and the clear finding of each experiment is not clearly written in individual sections which makes it even more difficult for the reader to understand the paper or the data presented here. 

L248: This is an important finding because the mortalities of these treatments (of what?) without food present at both 5 mM and with no ascorbic acid lay between 10-20% (of what?) at 1.000 mg/L (of what?).

L- 256 :This effect (?) of food presence (on what?) has been previously reported, and it reveals the dependence of the ingested dose (on what?) as regards food intake.

The numbering of individual sections are confusing.... for example L268 I was confused by the oscillation from 1.1 to 3.2.xxxx

L123, 137, 145 222; how can all the different sections be 1.1?

"The presence of food only during pre-exposure (PA) lowered mortality, but when food was administrated in the two phases (PP), it was similar to the total fast case (AA)." It is unclear what the authors really mean by two phases and the fast case. The administration procedure needs to be more clear.

L107-114: Please clarify if the nitrate was included as a part of the food or was it separately supplemented/provided to the nematodes.

Nematodes biosynthesize Ascorbic acid, please indicate if this information was factored in while determining the concentration of ascorbic acid treatment. Furthermore, is that supposed to effect the uptake/absorption of exogenous ascorbic acid? (https://www.ncbi.nlm.nih.gov/pmc/articles/PMC4357563/).

This result was in accordance with the previously observed reduction 299 in mortality, which explained the increase in ACT (Reference?)

A study in 2013 showed that nitric oxide increases the life span of c. elegans. I was just wondering if since both NO and nitrate (the subject of this study) are the sources of nitrogen, how can the authors reconcile these contradictory findings where NO appears to be supporting life span while NO3 is detrimental to it. (https://www.sciencedirect.com/science/article/pii/S0092867413000147)

Reviewer 2 Report

Authors have studied nitrite toxicity with the model Caenorhabditis elegans implementing a modeling approach. The manuscript is well written, however the authors must follow author instruction and revise the format and the reference of the entire manuscript

Abstract: please rewrite following author instructions

Please remove the Results from the material and methods Line 178, Line

Line 78-86  please rephrase is not clear the aim of the work

Line 79 C.elegans put in italics

Line 91 seeded change with which had been seeded

Line 91: the reference urgently needs a good update. It is my knowledge that we have much more modern scientific articles referred to the methodologies. I couldn’t believe that the method is the same of 50 years ago.

Line 96: please correct the format of the reference following the guidelines

Line 97: please rephrase

Line 99: please correct the format of the reference following the guidelines

Line 103: please correct the format of the reference following the guidelines

Line 227: please rephrase in correct English

Line 228: please remove dot from b.w; In rats is 121g or mg ?

Line 233-234 : please remove dot from b.w. use the same way of quantification /day or per day

Line 251-255: the figure caption must be near the figure

Line 257: please correct the reference following author guidelines

Line 379: please correct the reference following author guidelines

Line 441,  453,  484, 489, 490 : please correct the reference following author guidelines

Line 492: please correct the reference following author guidelines

References

Revise the references following author guidelines